# Climatic drivers of *Verticillium dahliae* occurrence in Mediterranean olive-growing areas of southern Spain

**Juan M. Requena-Mullor**[1,2‡], **Jose Manuel García-Garrido**[3], **Pedro Antonio García**[4],
**Estefanía Rodríguez**[5‡]*

**1** Department of Biological Sciences, Boise State University, Boise, ID, United States of America,
**2** Andalusian Center for the Assessment and Monitoring of Global Change (CAESCG), University of Almería,
Almería, Spain, **3** Department of Soil Microbiology and Symbiotic Systems, EEZ-CSIC, Granada, Spain,
**4** Department of Statistics, O.R., University of Granada, Granada, Spain, **5** Sustainable Crop Protection,
Institute for Agricultural and Fisheries Research and Training (IFAPA), La Mojonera, Almería, Spain

‡ These authors share first authorship on this work.
* mestefania.rodriguez@juntadeandalucia.es

**Data Availability Statement:** All data are available from the GitHub repository (https://github.com/jmrmcode/Verticillium-wilt-Dataset.git).

## Abstract

Verticillium wilt, caused by the soil-borne fungus *Verticillium dahliae*, is one of the most harmful diseases in Mediterranean olive-growing areas. Although, the effects of both soil temperature and moisture on *V. dahliae* are well known, there is scant knowledge about what climatic drivers affect the occurrence of the pathogen on a large scale. Here, we investigate what climatic drivers determine *V. dahliae* occurrence in olive-growing areas in southern Spain. In order to bridge this gap in knowledge, a large-scale field survey was carried out to collect data on the occurrence of *V. dahliae* in 779 olive groves in Granada province. Forty models based on competing combinations of climatic variables were fitted and evaluated using information-theoretic methods. A model that included a multiplicative combination of seasonal and extreme climatic variables was found to be the most viable one. Isothermality and the seasonal distribution of precipitation were the most important variables influencing the occurrence of the pathogen. The isothermal effect was in turn modulated by the seasonality of rainfall, and this became less negative as seasonality increases. Thus, *V. dahliae* occurs more frequently in olive-growing areas where the day-night temperature oscillation is lower than the summer-winter one. We also found that irrigation reduced the influence of isothermality on occurrence. Our results demonstrate that long-term compound climatic factors rather than "primary" variables, such as annual trends, can better explain the spatial patterns of *V. dahliae* occurrence in Mediterranean, southern Spain. One important implication of our study is that appropriate irrigation management, when temperature oscillation approaches optimal conditions for *V. dahliae* to thrive, may reduce the appearance of symptoms in olive trees.

**Funding:** E. Rodríguez; Caja Rural Foundation of Granada; https://www.fundacioncrg.com/; NO.

**Competing interests:** The authors have declared that no competing interests exist.

## Introduction

*Verticillium dahlia*e Kleb. is a widespread soil-borne pathogen reported for many high-value crops in temperate zones [1, 2]. This pathogen is a highly polyphagous fungus that infects hundreds of dicots, including herbaceous and woody plants, causing, chlorosis, necrosis, stunting, vascular discolouration, and wilting [3, 4]. *V. dahliae* is of great concern in Mediterranean olive groves [5–7], where the olive species under cultivation (i.e., *Olea europaea* L. subsp. *europaea*) have crucial economic, social and environmental value, and therefore, the European Union's agricultural policy is mainly aimed at protecting it [8]. Verticillium wilt is considered to be the most damaging disease for olives in Mediterranean countries as it causes yield losses and tree mortality [5, 9]. This disease thrives especially in intensively-farmed olive groves in Andalusia (southern Spain) [10] which is the world's leading olive tree grower, producing 900,000 tons of olive oil and 380,000 tons of table olives per year from 1.5 million hectares of farm land [11]. In western Andalusia, Verticillium wilt affects about 38–39% of crops [12, 13], while in the province of Granada (eastern Andalusia) *V. dahliae* affects around 14% of crops [14].

Verticillium wilt symptoms are divided into two categories which are associated with its two recognized pathotypes (i.e., virulent forms): defoliating (D), which is potentially lethal to the infested plant, and non-defoliating (ND), which causes a slow decline and partial defoliation [15, 16]. The predominance of virulent forms in Andalusian olive groves has strong implications for managing the disease [10]. Current protective measures for olives combine chemical treatments, resistant or tolerant cultivars or rootstocks, biological control, and physical and cultural solutions [5, 7, 10, 17–19]. However, this integrated strategy has not proved fully successful, as the pathogen has a wide range of hosts, and *V. dahliae* produces resting structures, i.e., microsclerotia, capable of surviving even when there is no host in the soil for years [20]. In fact, inoculum density in the soil is the main factor behind the spread of the disease in olive trees [21].

Verticillium wilt development is driven by several factors such as host plant, inoculum density, pathogen dispersal mechanisms, and environmental conditions [1, 22]. The latter may, in turn, be influenced by climate change [23]. Specifically, the spread of the disease depends on inoculum density in the soil, which is strongly influenced by soil temperature and seasonal fluctuations [24, 25]. Verticillium wilt is mainly observed in the spring, whereas it decreases in the summer due to high temperatures [9, 12, 26–31]. Previous studies have identified some abiotic factors behind the occurrence of the pathogen in olive groves in the Granada province [14, 32, 33]. These authors found that *V. dahliae* was up to 3 times more prevalent in irrigated olive groves than rainfed ones, and was more frequent when plant material came from nurseries than when it did not. In fact, the prevalence of the pathogen in olive plantations whose plant material came from nurseries reached 24.5%. Therefore, these findings highlight the importance of using cuttings which have been certified as free of the pathogen, and the role the infected plant material has in spreading the disease to non-infested cropping areas. Additionally, slope gradients and soil type both influence occurrence of the pathogen. For instance, *V. dahliae* occurrence was higher in olive groves located in saline, alkaline and steep slope soils [33].

The effects of both soil temperature and moisture on *V. dahliae* are well known locally (i.e. plot scale). For instance, Verticillium wilt in olive is favored by soil temperatures between 20 – 25˚C [1]. The D pathotype is promoted by soil temperatures, ranging from 16 to 24˚C whereas the ND pathotype thrived in lower soil temperatures, ranging from 16 to 20˚C [25]. Similarly, several studies have shown that disease development, severity and inoculum density in olive fields are usually higher in wetter soils [5, 34, 35]. However, there is scant knowledge

about how climatic factors affect the occurrence of the pathogen on a large scale. In fact, fungal plant pathogens are particularly under-represented in climatic niche modeling studies [36] but see [37]. Bridging this gap in knowledge remains a major challenge in research, partly due to the absence of spatially explicit data on the occurrence of *V. dahliae* on a larger scale. One notable study is that by [38] on the diversity of the pathogen population in southern Spain. Identifying the climatic drivers behind the occurrence of *V. dahliae* on a large scale will provide an insight into the pathogen ecology and help optimize strategies for managing the disease in olive groves.

Here, we used the data set collected by [14, 32, 33] in the Granada province to determine by modelling what climatic factors affect *V. dahliae* occurrence in Mediterranean olive-growing areas of southern Spain. To this end, we tested a series of models with different combinations of climatic variables representing competing hypotheses, and used information-theoretic methods to select the most parsimonious model [39].

## Materials and methods

The sampling survey, isolation and *V. dahliae* identification protocols used in this research are based on previous studies [14, 32, 33].

### Study area and sampling survey design

There are 183,000 ha of olive groves in the province of Granada, making it the third largest olive oil producer in Andalusia [40]. In this province, olives are distributed among olive-growing areas which vary greatly in terms of their physical characteristics, management practices, olive varieties and environmental factors [14]. Thus, a stratified double-sampling technique was designed in order to select the olive groves [14]. This technique is commonly applied to heterogeneous populations and collecting data based on sample sizes that are proportional to the relative size of the strata [41]. Proportions in the first sampling level were based on the surface area (ha) of olive groves in each olive-growing zone. The second level consisted of olive groves whose probability of infection was proportional to their size (PPS). PPS gives a probability of selecting a sampling unit (e.g., olive grove) in proportion to the size of its population [41]. It was assumed a priori that the probability of an olive grove being infected with *V. dahliae* was 50%. Finally, the sampling survey for evaluating occurrence of *V. dahliae* in the study area was performed in 779 olive groves (Fig 1), over 2,833 ha with 139 olive trees/ha and a standard error of $\leq 2.48$ [14].

### Isolation and identification of *V. dahliae* in olive trees

From 2003 to 2005, a sampling survey was carried out during the spring and autumn, the most suitable periods for assessing the occurrence of *V. dahliae* in plants [29]. Wherever possible, ten olive trees with typical symptoms of *V. dahliae* were sampled for each olive-grove [14]. Verticillium isolates from olive trees were recovered from twig tissues using the standardized isolation technique [42]. For each tree, six olive twigs were washed thoroughly under running tap water and debarked. 5 mm long olive twigs sections were taken from a laminar flow bench, whose surfaces were sterilized in 10% sodium hypochlorite for 1 min and rinsed twice with sterile distilled $H_2O$ and dried on sterilized paper. After drying completely, eight twig sections were transferred to 90 mm Petri plates with water agar amended with Aureomycin (1 l of distilled water, 20 g of agar and 30 mg of aureomycin). Three Petri plates were used per tree. All samples were incubated at 25°C in the dark for 30 days [14, 32, 33]. *V. dahliae* isolates were initially identified on the basis of morphological characteristics of verticillate conidiophores and the formation of microsclerotia.

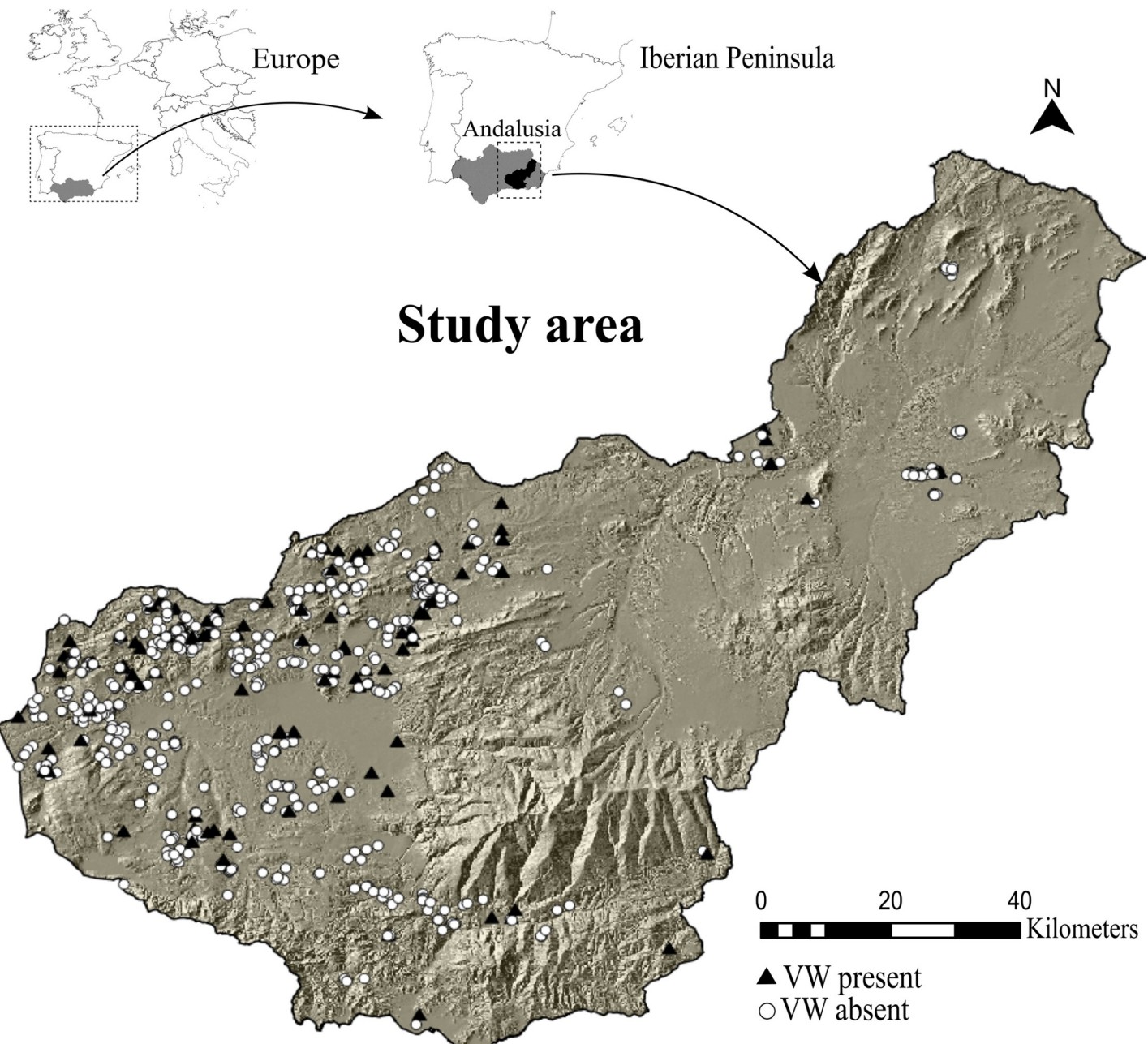

**Fig 1. The Granada province study area boundary and the location of the 779 survey records that comprised the presence and absence of point locations for *V. dahliae* used in this study (see Materials and Methods section).** The administrative boundaries were obtained from public spatial data layers of the Environmental Information Network of Andalusia (https://descargasrediam.cica.es/repo/s/RUR?path=%2F). The digital elevation model showed as a base map was downloaded from a public database available in (http://www.juntadeandalucia.es/institutodeestadisticaycartografia/prodCartografia/bc/mdt.htm). Date accessed: October 1st, 2019. All the data is compatible with the CC-BY 4.0 license.

Fungal DNA was extracted using the DNeasy Plant Kit (Qiagen, Germany) from harvested mycelia of *V. dahliae* isolates growing in PDB (Potato Dextrose Broth) [14, 32, 33]. DNA was amplified in PCR assays to confirm the identity of the 123 *V. dahliae* isolates recovered. Primers DB19/DB22 amplified *V. dahliae* specific DNA bands of 523 or 539 bp [43].

## Environmental variables

We checked the correlations between the nineteen available bioclimatic variables in the WorldClim version 2 database (1970–2000) (http://www.worldclim.org/) and selected the five with a Spearman rank-correlation under 0.6 [44], i.e., annual rainfall (AnnualRainfall), intra-annual rainfall seasonality (coefficient of variation) (RainfallSeasonality), intra-annual temperature seasonality (standard deviation x 100) (TempSeasonality), maximum temperature of the warmest month (MaxTemp), and isothermality ((mean diurnal temperature range/temperature annual range) x 100) (Isothermality). Isothermality quantifies the degree to which day-to-night temperatures oscillate in relation to summer-to-winter (annual) oscillations [45] (see S1 Fig). These variables were downloaded in BIL/HDR format at a cell size of 30 arc-seconds (~1 km) using the R package "raster" [46]. We found the average of the climatic variables using a buffer based on the area and centroid of each olive grove, both of which were obtained from the Spanish Ministry for Agriculture, Fisheries and Food (http://www.magrama.gob.es/es/). Similarly, both watering (i.e., irrigated vs. rainfed) and plant material origin (i.e., annual pruning (traditional propagation of large hard-wood cuttings) vs. young plants from nurseries) were included as covariates in the models. Irrigation was shown to influence the occurrence of *V. dahliae* in the study area [14]. Likewise, the origin of the propagation material also had a bearing on this [33]. Finally, since plantation size was expected to positively affect pathogen occurrence, the olive grove area (in ha) was also included as a covariate. The data set is available at https://github.com/jmrmcode/Verticillium-wilt-Dataset [47].

## Modelling approach

Although pathogens are mainly dependent on the occurrence of their hosts, climate is also an important factor that influences their distribution on a wider scale [48]. Indeed, complex interactions among multiple climatic factors influenced the distribution of a range of plant pathogenic fungi commonly reported in Europe [49–52]. Here, we explored the climatic drivers which affected *V. dahliae* occurrence by testing a series of competing models with different combinations of climatic variables associated with five hypotheses and then, selected the most parsimonious model using information-theoretic methods (see below) [39]. Specifically, we designed a set of forty candidate Generalized Linear Models (GLM) grouped into the following hypotheses: *V. dahliae* occurrence on a large scale depended on (1) annual trends in climatic variables, (2) seasonality of climatic variables, (3) extreme climatic variables, (4) an additive combination of hypotheses 1, 2, and 3, and (5) an additive and multiplicative (interactions) combination. We showed occurrence of *V. dahliae* by allocating a "1" to those olive groves where the detection was positive and "0" when it was negative (Fig 1). All predictor variables were standardized to have a mean of 0 and a standard deviation of 1. We applied the complementary log-log (cloglog) function Eq (1) to link *V. dahliae* occurrence and the linear combination of predictors and covariates. The cloglog function is not symmetric in the interval [0,1] and therefore, it is frequently used for binary responses when the probability of an event is very low [53].

$$Cloglog(x) = log(-log(1-\pi)) \tag{1}$$

where $\pi$ is the probability of *V. dahliae* occurrence. The information-theoretic approach was used to select the most parsimonious model/s and rank the remaining ones [54]. The Akaike information criterion (AIC) is a measure of model fit that penalizes the likelihood according to the number of parameters in the model. The lower the AIC value, the more parsimonious the model is [54]. Delta AIC (ΔAIC) was calculated as the difference in AIC between each model and the lowest AIC value in the series. Models with ΔAIC < 2 have "considerable"

support [54]. In addition, Akaike weights ($w_i$) were calculated to rank models with $\Delta$AIC < 2, in turn. In order to verify whether the interactions between climatic and abiotic factors were significant, we included interaction terms between climatic variables, watering and plant material origin in the top-ranked model (see S1 Table for details).

We assessed how accurate the predictions of the model were by comparing the Log-Loss [55] achieved by the null-model (i.e., intercept-only) as a baseline with that of the top-ranked model using a 10-fold cross-validation of the presence-absence sampled data. Log-Loss measures the uncertainty of fitted probabilities by comparing them to actual observations. Each data partition was made by randomly splitting the data into 10 distinct blocks and using 9 to train and 1 to test. A key assumption for regression is that model residuals were not correlated, we checked this assumption by estimating the Moran's I statistic using the inverse of the distances between observations as weights [56]. All statistical analyses were performed with the R software V. 3.6.2 (R Core Team 2019) using the "qpcR", "spdep", "Metrics", and "stats" packages [56–59]. Finally, we carried out a sensitivity analysis to assess how much models performance changes when using a different modeling technique. To do that, we compared the prediction capacity of our top-ranked logistic model (see Table 1) with boosted regression trees using a 10-fold cross-validation (see S1 Appendix for details).

According to the rule that $\Delta$AIC < 2 suggests the best parsimony in a group of candidate models, two models (marked in bold) was selected to likely explain occurrence of *V. dahliae* in the study area. AIC: Akaike Information Criterion; $\Delta$AIC: AIC differences; $w_i$: Akaike weights.

## Results

*V. dahliae* was detected in 122 of the 779 olive groves surveyed, yielding a sample occurrence of 0.17, which justified the use of cloglog as the link function in our models. Two models reached the highest parsimony rank according to AIC scores criterion, i.e., $\Delta$AIC < 2 (Table 1). These models consisted of a combination of seasonal (rainfall seasonality) and extreme climatic variables (isothermality) (Hypotheses 4 and 5). Isothermality had a negative effect on *V. dahliae*, with a predictive decrease in occurrence of 53.9% (*p-value* = 0.0006) when this variable increased by 0.45% in rainfed olive groves. However, rainfall seasonality had no discernable marginal effect on pathogen occurrence (Fig 2). The top-ranked model included an interaction term between both climatic variables. We found that when rainfall seasonality increased by 4.4 (i.e., standard deviation of 1) the effect of isothermality on *V. dahliae* occurrence rose by 86.5% (*p-value* = 0.04) (Figs 2 and 3), and therefore, this effect became less negative. Considering the relative importance of the interaction between climatic factors and watering and plant material origin in the top-ranked model, we found that by including the interaction between isothermality and watering, the overall model performance improved (AIC = 559.8) and there was a significant effect on *V. dahliae* occurrence (Fig 2 and S1 Table). As olive groves changed from rainfed to irrigated, the effect of isothermality on occurrence became less negative by 66.8% (*p-value* = 0.04) (Fig 3).

As expected, the predictive capacity of the top-ranked model was better than that of the intercept-only model (mean Log-Loss = 0.36 ±0.013 SE and 0.43 ±0.015 SE, respectively; lower Log-Loss values indicate better performance), with an improvement in the predictions of 22.24% ±12.64 SE (see S2 Table). No spatial patterns were found in the model residuals (Moran's I statistic = 0.027; p-value = 0.008; see S2 Fig) and this indicated that the climatic variables included in the top-ranked model did indeed account for the spatial variation of *V. dahliae* occurrence. Finally, the model comparison results showed no evidence of improving predictive performance by the BRT models compared to that obtained by the top-ranked binomial model, supporting the linear regression modeling approach adopted in our study (see S3 Table in S1 Appendix).

**Table 1. Highest ranked competing models for *Verticillium dahliae* occurrence in olive groves of Granada province (southern Spain).**

| Ranking | Model | AIC | ΔAIC | w$_i$ |
|---|---|---|---|---|
| **1** | **RainfallSeasonality x Isothermality** | **562.83** | **0.00** | **0.273** |
| **2** | **RainfallSeasonality + Isothermality** | **564.29** | **1.46** | **0.132** |
| 3 | AnnualRainfall x RainfallSeasonality x TempSeasonality | 565.23 | 2.4 | 0.082 |
| 4 | RainfallSeasonality x TempSeasonality x Isothermality | 565.71 | 2.88 | 0.065 |
| 5 | RainfallSeasonality + TempSeasonality + Isothermality | 565.95 | 3.12 | 0.057 |
| 6 | RainfallSeasonality + MaxTemp + Isothermality | 565.99 | 3.16 | 0.056 |
| 7 | MaxTemp + Isothermality | 566.62 | 3.79 | 0.041 |
| 8 | Isothermality | 566.67 | 3.84 | 0.040 |
| 9 | RainfallSeasonality x MaxTemp x Isothermality | 567.51 | 4.68 | 0.026 |
| 10 | RainfallSeasonality + TempSeasonality + MaxTemp + Isothermality | 567.89 | 5.06 | 0.022 |
| 11 | TempSeasonality + MaxTemp + Isothermality | 567.92 | 5.09 | 0.021 |
| 12 | TempSeasonality x MaxTemp x Isothermality | 567.94 | 5.11 | 0.021 |
| 13 | AnnualRainfall + MaxTemp + Isothermality | 568.15 | 5.32 | 0.019 |
| 14 | TempSeasonality x Isothermality | 568.35 | 5.52 | 0.017 |
| 15 | AnnualRainfall x Isothermality | 568.36 | 5.53 | 0.017 |
| 16 | RainfallSeasonality x TempSeasonality x MaxTemp x Isothermality | 568.41 | 5.58 | 0.017 |
| 17 | TempSeasonality + Isothermality | 568.47 | 5.64 | 0.016 |
| 18 | AnnualRainfall + Isothermality | 568.55 | 5.72 | 0.016 |
| 19 | RainfallSeasonality x MaxTemp | 569.02 | 6.19 | 0.012 |
| 20 | AnnualRainfall + RainfallSeasonality + TempSeasonality + MaxTemp + Isothermality | 569.87 | 7.04 | 0.008 |
| 21 | AnnualRainfall + MaxTemp | 570.36 | 7.53 | 0.006 |
| 22 | MaxTemp | 570.47 | 7.64 | 0.007 |
| 23 | AnnualRainfall x MaxTemp | 570.56 | 7.73 | 0.006 |
| 24 | RainfallSeasonality + TempSeasonality | 571.79 | 8.96 | 0.003 |
| 25 | RainfallSeasonality + MaxTemp | 572.14 | 9.31 | 0.003 |
| 26 | AnnualRainfall x RainfallSeasonality x TempSeasonality x MaxTemp x Isothermality | 572.19 | 9.36 | 0.003 |
| 27 | TempSeasonality + MaxTemp | 572.29 | 9.46 | 0.002 |
| 28 | RainfallSeasonality x TempSeasonality x MaxTemp | 572.31 | 9.48 | 0.002 |
| 29 | AnnualRainfall x MaxTemp x Isothermality | 572.54 | 9.71 | 0.002 |
| 30 | RainfallSeasonality + TempSeasonality + MaxTemp | 572.67 | 9.84 | 0.002 |
| 31 | TempSeasonality x MaxTemp | 572.92 | 10.09 | 0.002 |
| 32 | AnnualRainfall + RainfallSeasonality + TempSeasonality | 573.29 | 10.46 | 0.001 |
| 33 | TempSeasonality | 577.8 | 14.97 | 1.5E-04 |
| 34 | RainfallSeasonality | 579.4 | 16.57 | 6.9E-05 |
| 35 | AnnualRainfall + TempSeasonality | 579.76 | 16.93 | 5.8E-05 |
| 36 | AnnualRainfall + RainfallSeasonality | 580.22 | 17.39 | 4.6E-05 |
| 37 | AnnualRainfall x TempSeasonality | 580.46 | 17.63 | 4.0E-05 |
| 38 | AnnualRainfall | 580.53 | 17.7 | 3.9E-05 |
| 39 | AnnualRainfall x RainfallSeasonality | 581.4 | 18.57 | 2.5E-05 |
| 40 | Intercept only | 678.18 | 115.35 | 2.5E-26 |

## Discussion

*V. dahliae* is one of the most harmful pathogens in the Mediterranean region [60], but, surprisingly, no studies have identified what climatic drivers determine the occurrence of this pathogen on a large scale. We found that long-term compound climatic factors rather than 'primary' variables, such as annual trends, provided a more comprehensive explanation of the spatial patterns of *V. dahliae* occurrence in olive-growing areas in southern Spain. Here, compound

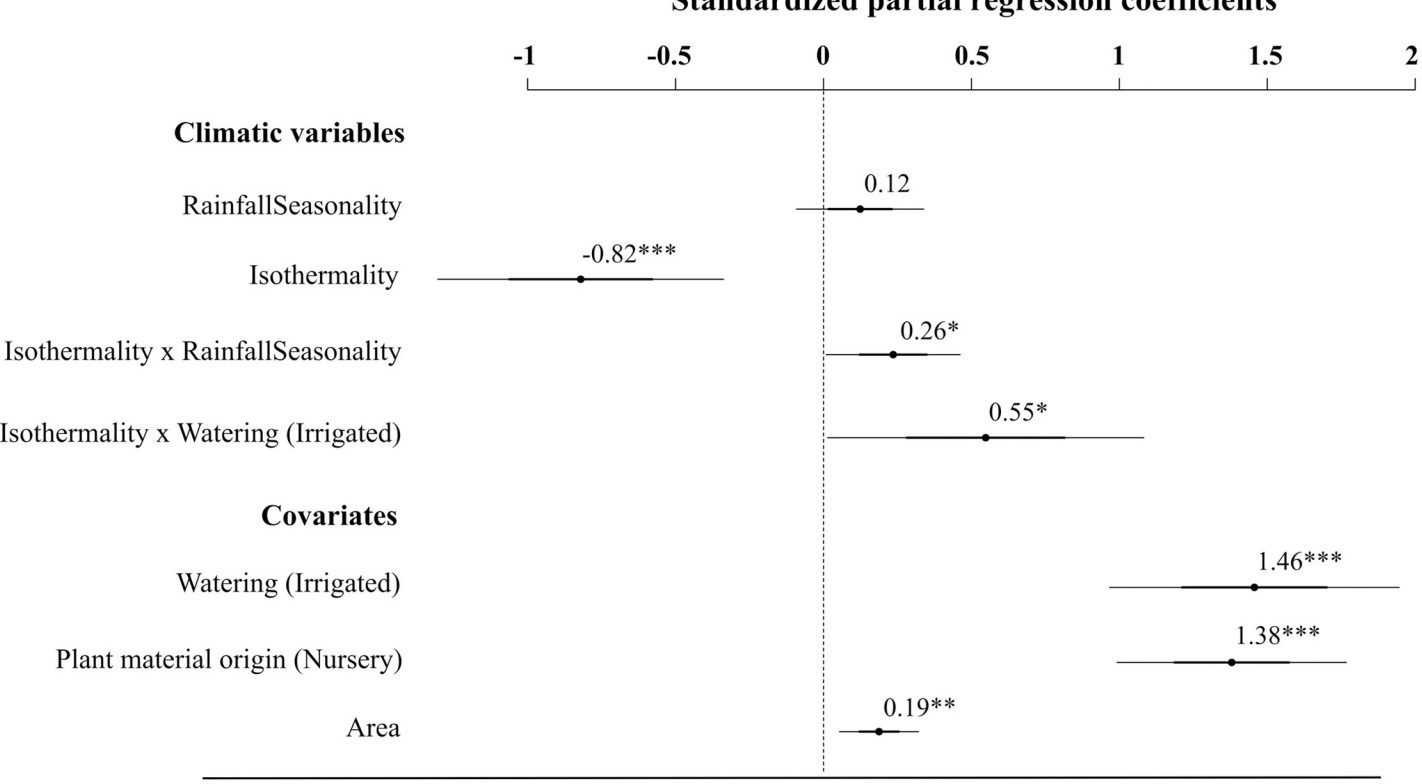

**Fig 2. Effect size plot of the climatic variables and abiotic covariates influence on *Verticillium dahliae* occurrence according to the most plausible model (see Table 1).** Effect sizes are shown on a cloglog scale. Rainfed and annual pruning were treated as reference levels for watering and plant material origin, respectively.

climatic factors refer to climatic variables that integrate different aspects of climate components at multiple temporal scales. For example, isothermality represents the ratio between temperature oscillations at two temporal scales: daily and seasonal, and it captures wider ecological trends better than the temperature for a given day (i.e., a "primary" variable) due to the inherent variability associated with weather. Isothermality was the most important climatic factor for the pathogen occurrence, suggesting an intricate relationship between *V. dahliae* and climate as discussed below.

In the Western Mediterranean basin, seasonal rainfall regimes are equinoctial, with a rainy season in autumn and another secondary rainy period in spring [61]. In contrast, maximum and minimum temperatures are reached in summer and winter, respectively. Specifically, in Granada province, precipitation mainly occurs in November and early spring, while there is almost no precipitation in July and August. Summers are extremely hot and winters moderately cold, with large variations in temperature between the day and night (maximum temperature > 30˚C; minimum temperature < 20˚C) [62]. Thus, high rainfall seasonality and isothermality, i.e., the two climatic factors that determined Verticillium wilt occurrence in Granada province, implies more varied distribution of rainfall throughout seasons, as well as greater differences between cold and hot periods, with mild temperatures in between. Our results, confirmed findings from previous reports, by demonstrating that weather oscillation helped *V. dahliae* thrive in Mediterranean olive groves, i.e., areas where rainfall mainly occurred in autumn and early spring, and with mild temperatures [6, 35]. According to [6], the optimal temperature range for *V. dahliae* to thrive is between 22˚-25˚C. The optimum

**(a)** **Isothermality x RainfallSeasonality**

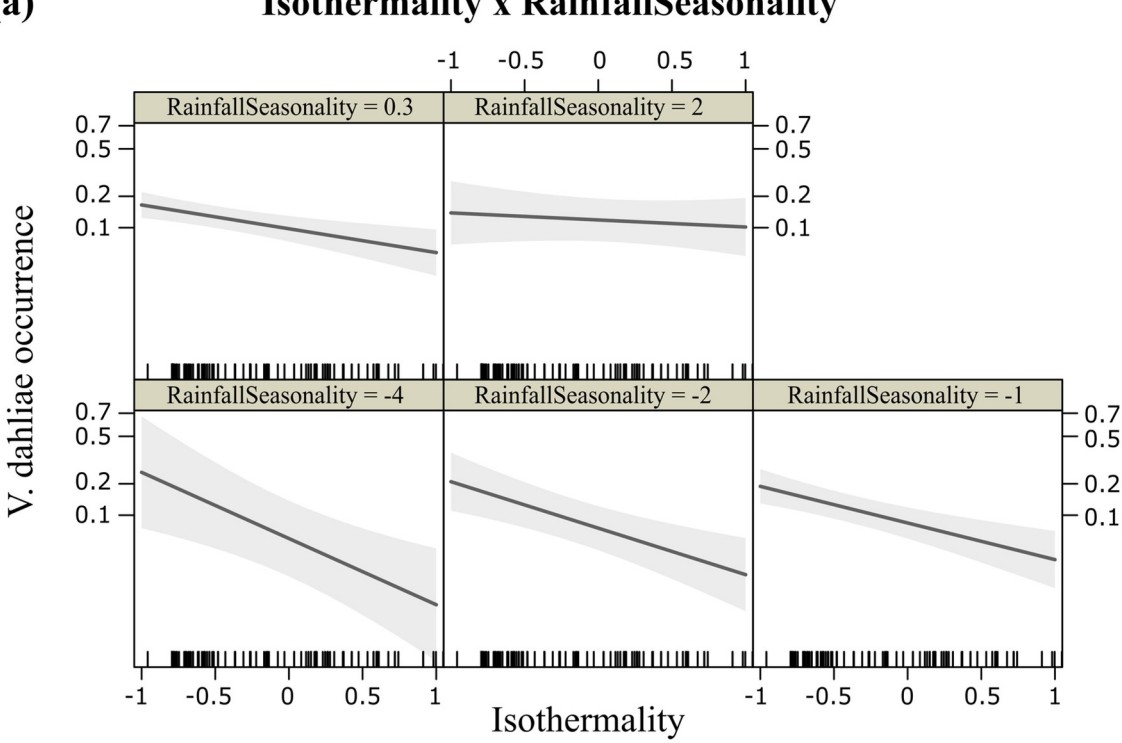

**(b)** **Isothermality x Watering**

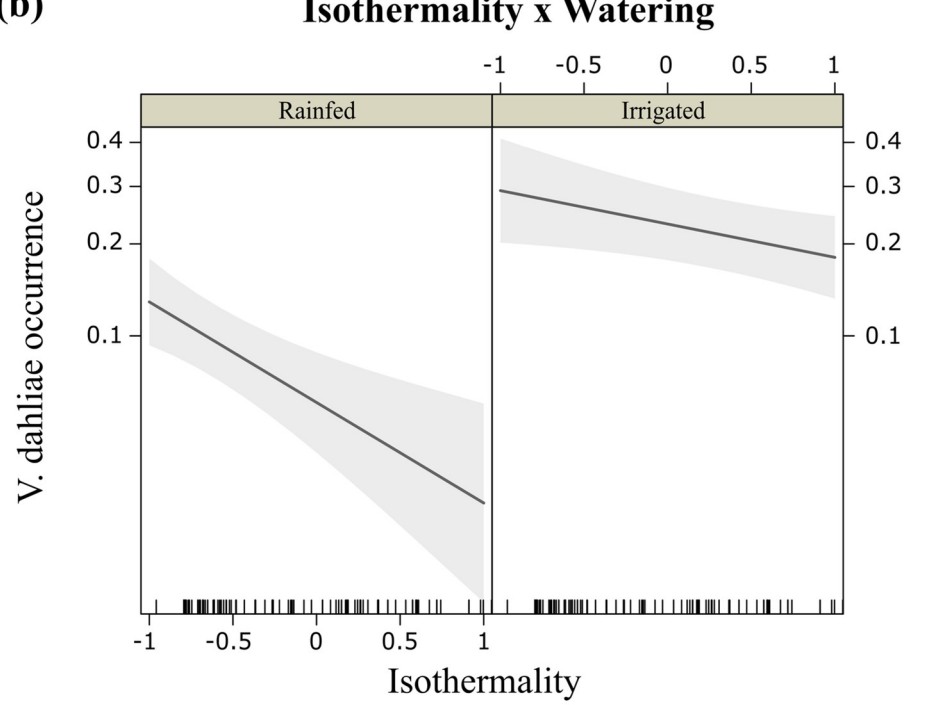

**Legend**: symbol "x" stands for interaction. **(a)** From top to bottom and from right to left, the facets show the effect of isothermality on *V. dahliae* occurrence as the seasonality of rainfall increases. **(b)** The effect of isothermality on *V. dahliae* occurrence in rainfed (left) and irrigated olive groves (right).

**Fig 3.** The predicted effect of Isothermality on *Verticillium dahliae* occurrence as a function of **(a)** RainfallSeasonality and **(b)** Watering according to the most plausible model (see Table 1). Effect sizes are shown on a cloglog scale. Shaded areas represent the 95% confidence interval for the fitted cloglog curve.

temperature of the soil for *V. dahliae* was around 24˚C, after which there was a decline in the abundance of the pathogen when soil temperature exceeded 28˚C [25]. Several authors observed that a high temperature in summer can suppress *V. dahliae* in olive trees [29, 30, 63].

We also corroborated previous results that indicated irrigation was a significant abiotic factor for *V. dahliae* [14], since there was a strong positive correlation between irrigated olive groves and *V. dahliae* occurrence. Furthermore, we provided new findings on the relationship between climate and watering by showing how irrigation can mitigate the negative effect of isothermality on occurrence. Many studies have shown a relationship between *V. dahliae* and water abundance in soil. The inoculum of the pathogen was more frequently found in moist soil around drippers [34]. In addition, the disease incidence was positively associated with the number of years of irrigation and water dosage [64]. The damage caused by *V. dahliae* in Mediterranean olive groves has steadily increased over the last few decades due to the implementation of irrigation systems [12, 65, 66], which may help propagules of the pathogen to disperse [67, 68]. A similar process occurs in cotton, which is not affected by Verticillium wilt under rainfed conditions, although it is with irrigation [20]. In accordance with previous results, we propose that water from irrigation may keep soil temperature close to the optimal range for *V. dahliae*, and hence help it survive in areas where day-tonight temperature oscillations are higher. Regarding the marginal effect of the plant material origin, our results were also consistent with that found by [33] in which young plants from nurseries were found to increase the probability of *V. dahliae* occurrence. This result underlines the importance of abiotic factors in reducing disease risk during the early stages of cultivation. Infected planting material from the nursery is one of the main reasons for *V. dahliae* infestation in free-pathogen growing areas [33]. In order to prevent the pathogen from spreading, particularly the most virulent forms, such as the defoliating pathotype (D), farmers should use only pathogen-tested propagative material. Since the role of non-climatic factors on the pathogen is beyond our study, we recommend reading the above mentioned article for further information.

We expected annual precipitation to be a significant factor affecting the occurrence of the pathogen in olive groves. However, we found that models that included rainfall seasonality had a better fit. This finding may be due to the fact that optimal soil moisture and temperature conditions are required to occur simultaneously. Rainfall in semi-arid areas, such as around Granada, is scarce and irregularly distributed in time and space [69–71]. Since *V. dahliae* thrives in mild temperature conditions [6], rainy periods must coincide with warmer ones. These findings highlight the seasonal nature of the pathogen, as reported in other studies on Mediterranean olive growing-areas, which found *V. dahliae* caused wilt disease in olive trees in late-spring and early-winter [9, 26, 27, 29–31]. Moreover, our results showed that rainfall seasonality modulated the negative effect of isothermality, which became more negative as the seasonality decreased. We propose that the high occurrence of the pathogen found in the northwest of Granada province [14] can be explained by the combination of low rainfall seasonality and high isothermality that characterizes such area.

Our findings have important implications for controlling the spread of *V. dahliae* in semi-arid Mediterranean olive-growing areas, as the risk of Verticillium wilt can be reduced by managing irrigation. Since irrigation may reduce the negative influence of isothermality on the pathogen occurrence, shifting from rainfed to irrigated olive groves may entails a higher risk in those areas with low isothermality. This combination draws areas potentially susceptible to *V. dahliae* occurrence, for example, the irrigated growing-olive areas located in the north of

the study area which have low isothermality and the highest occurrence of Verticillium wilt in Granada province [14]. In contrast, the low pathogen's occurrence found in the south [14] may be a result of the conjunction of rainfed olive grove and high isothermality. We showed that *V. dahliae* is a highly seasonal pathogen that requires rainy and warm periods to occur simultaneously. On this basis, reducing irrigation in spring and autumn could prevent the disease from spreading, particularly in areas with low rainfall seasonality and isothermality. Our results concur with that of [35], who suggested that scheduling irrigation treatment on the basis of rainfall could be a good strategy for maintaining soil moisture below levels which are favourable to the pathogen. They concluded that daily -as opposed to both weekly and biweekly- irrigation helps Verticillium wilt thrives. They also advised that in order to reduce symptoms of the disease, soil water content should be lower than 24% and irrigation should be done on a biweekly basis. However, it should be stressed that although less frequent irrigation could reduce the risk of Verticillium wilt, this will need to be balanced with the water needs of the olive trees in order to achieve good and high-quality oil yields. For instance, irrigation prior to anthesis produced lower fusarium head blight than continuous irrigation throughout anthesis [72]. However, finishing irrigation prior to anthesis resulted in reduced yields in comparison with continuous irrigation. Water stress from inflorescence development (mid-April to the start of June) to fruit set (May to June) should be avoided. Also, the periods of initial fruit growth and oil accumulation in the autumn are sensitive to water deficits (September to October) [73]. Consequently, there are 3 extremely critical periods for crop productivity, during which time soil moisture must be kept optimal. Unfortunately, these periods also coincide with Verticilllium wilt development. Therefore, studies need to be made to determine disease development, soil moisture levels and rainfall during those critical periods to assess influences on the pathogen and olive tree yields. Evidentially the response of any given soil-borne disease to the range of ways of managing irrigation varies widely and must be addressed for each individual plant-pathogen system [74].

## Conclusions

Our study is the first to explore what climatic factors influence the occurrence of *V. dahliae* in Mediterranean olive groves in southern Spain. Isothermality had a negative effect on occurrence, but this was modulated by rainfall seasonality and watering. Since *V. dahliae* thrives in mild temperature conditions, rainy periods must coincide with warm ones in order to provide optimal conditions for the pathogen to thrive. Therefore, appropriate irrigation management in periods when the pathogen spreads, i.e., spring and autumn, may prevent symptoms of the disease. However, reducing irrigation during spring and autumn may also have undesirable side effects such as decreased yields. Thus, future studies at finer scales that consider microclimate conditions in periods in which Verticillium wilt develop are required to assess how to best manage irrigation.

## Supporting information

**S1 Table. Comparison of model fit when including interaction terms between climatic variables and covariates.** Each pairwise interaction was separately included in the top-ranked model for *V. dahliae* occurrence (see Table 1 in the main manuscript). We ordered the models by their AIC values and checked whether the interaction term was significant or not. (DOCX)

**S2 Table. Comparison between the top-ranked model and the intercept-only model using 10-fold cross-validation.** The performance of the models was evaluated using the Log-Loss. Lower values of this model selection index indicate better model performance. Performance

measures were averaged throughout the ten data partitions. Improvement was calculated as [(Top-ranked—Only-intercept) / Only-intercept] * 100. Each data partition was made by randomly splitting the data into 10 distinct blocks and using 9 to train and 1 to test.
(DOCX)

**S1 Fig. Map of WorldClim climatic variables averaged for the years 1970–2000 in Granada province (southern Spain).**
(DOCX)

**S2 Fig. Analysis of the top-ranked model residuals.** (a) Bubble plot of the standardized residuals from the top-ranked model (see Table 1 in the main manuscript) including an interaction term between isothermality and watering. (b) Spline correlogram of the residuals using the function *spline.correlog* in the "*ncf*" R package [1]. The spatial dependence is tested as a continuous function of distance. The gray shadows represent the 95% confidence interval.
(DOCX)

**S1 Appendix. Comparison between boosted regression trees and binomial models for *V. dahliae* occurrence.**
(DOCX)

## Acknowledgments

We would like to thank farmers from Granada province for kindly permitting us to research their olive groves. We also wish to give special thanks to Mr. Antonio León, chief executive officer of Caja Rural in Granada, for encouraging this research and Toby Wakely for reviewing this manuscript in English.

## Author Contributions

**Conceptualization:** Juan M. Requena-Mullor, Jose Manuel García-Garrido, Pedro Antonio García, Estefanía Rodríguez.

**Data curation:** Estefanía Rodríguez.

**Formal analysis:** Juan M. Requena-Mullor, Estefanía Rodríguez.

**Funding acquisition:** Jose Manuel García-Garrido, Pedro Antonio García, Estefanía Rodríguez.

**Methodology:** Juan M. Requena-Mullor, Jose Manuel García-Garrido, Pedro Antonio García, Estefanía Rodríguez.

**Project administration:** Estefanía Rodríguez.

**Supervision:** Jose Manuel García-Garrido, Pedro Antonio García, Estefanía Rodríguez.

**Validation:** Juan M. Requena-Mullor, Jose Manuel García-Garrido, Pedro Antonio García, Estefanía Rodríguez.

**Writing – original draft:** Juan M. Requena-Mullor, Estefanía Rodríguez.

**Writing – review & editing:** Juan M. Requena-Mullor, Estefanía Rodríguez.

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
