## [Decision Letter · Decision Letter 0]

15 Jul 2020

PONE-D-20-11197

Climatic drivers of Verticillium dahliae occurrence in Mediterranean olive-growing areas of southern Spain

PLOS ONE

Dear Dr. Rodríguez,

Thank you for submitting your manuscript to PLOS ONE. After careful consideration, we feel that it has merit but does not fully meet PLOS ONE’s publication criteria as it currently stands. Therefore, we invite you to submit a revised version of the manuscript that addresses the points raised during the review process.

Both referees are overall positive about the manuscript, although some important concerns have been raised about specific methodological aspects and the discussion of the main findings. Specifically, one of the referee suggested to explore more sophisticated modelling techniques (e.g. Boosted regression trees) to account for covariates interactions. In addition, the choice of environmental predictors has been judged inappropriate with respect of the spatial scale of the study. Even more importantly, the link between study results and species ecology was not adequately emphasized and discussed. I strongly suggest to take comprehensively into account such valuable suggestions, which can surely improve the overall quality of the study.

We look forward to receiving your revised manuscript.

Kind regards,

Mirko Di Febbraro

Academic Editor

PLOS ONE

Journal Requirements:

3. We note that Figure 1 in your submission contain map images which may be copyrighted. All PLOS content is published under the Creative Commons Attribution License (CC BY 4.0), which means that the manuscript, images, and Supporting Information files will be freely available online, and any third party is permitted to access, download, copy, distribute, and use these materials in any way, even commercially, with proper attribution. For these reasons, we cannot publish previously copyrighted maps or satellite images created using proprietary data, such as Google software (Google Maps, Street View, and Earth). For more information, see our copyright guidelines: http://journals.plos.org/plosone/s/licenses-and-copyright.

3.1.    You may seek permission from the original copyright holder of Figure 1 to publish the content specifically under the CC BY 4.0 license.

3.2.    If you are unable to obtain permission from the original copyright holder to publish these figures under the CC BY 4.0 license or if the copyright holder’s requirements are incompatible with the CC BY 4.0 license, please either i) remove the figure or ii) supply a replacement figure that complies with the CC BY 4.0 license. Please check copyright information on all replacement figures and update the figure caption with source information. If applicable, please specify in the figure caption text when a figure is similar but not identical to the original image and is therefore for illustrative purposes only.

Reviewers' comments:

Reviewer's Responses to Questions

**Comments to the Author**

1. Is the manuscript technically sound, and do the data support the conclusions?

Reviewer #1: Yes

Reviewer #2: Partly

2. Has the statistical analysis been performed appropriately and rigorously? 

Reviewer #1: Yes

Reviewer #2: Yes

3. Have the authors made all data underlying the findings in their manuscript fully available?

Reviewer #1: Yes

Reviewer #2: Yes

4. Is the manuscript presented in an intelligible fashion and written in standard English?

Reviewer #1: Yes

Reviewer #2: Yes

5. Review Comments to the Author

Reviewer #1: In this study I appreciated the attempt to investigate the correlation between climatic variables and the probability of occurrence V. dahliae pathogen by applying modelling and statistic approaches. Moreover, the results of your study may be very helpful to provide information about how irrigation could be better managed in order to modulate the effect of climate on fungus’s thrive and reduce the risk of pathogen’s spread.

Nevertheless, before I recommend this paper for publication, the authors should address some issues that I detected.

Introduction

- Line 47: I would write “in Andalusia (southern Spain) which is the world’s leading olive tree grower…”

Methods

- Line 139: A citation about the application of the Spearman rank-correlation is needed.

- Line 159: There are other studies that have investigated how climatic factors might affect the distribution of plant

pathogenic fungi, e.g.: Bosso, L., Luchi, N., Maresi, G., Cristinzio, G., Smeraldo, S., & Russo, D. (2017). Predicting current and future disease outbreaks of Diplodia sapinea shoot blight in Italy: species distribution models as a tool for forest management planning.

- Line 186: Please, clarify if you have used the Log Loss with presence-only test data.

- Line 198 and line 212: Please, remove citations from results session and eventually move them to discussion session.

- Lines 228-229: Authors should show the full results of log loss evaluation in supplementary materials.

Discussion

- Lines 236-237: Authors should specify why they use the term “sophisticated” and “primary” for the different variables.

- Line 248: Please, a citation is needed.

- Lines 257-259: Authors say that irrigation mitigate the negative effect of isothermality on V. dahliae occurrence, however, if I have correctly understood, from fig.2 it results that the combination between isothermality and watering showed an increasing negative effect on occurrences. Please, could you better explain this sentence?

Reviewer #2: Dear authors,

Thank you for the opportunity to review the manuscript titled ‘Climatic drivers of Verticillium dahliae occurrence in Mediterranean olive-growing areas of southern Spain’. This manuscript presents a straightforward study, using species distribution modelling techniques to interrogate conditions potentially favourable to the studied organism. The methods used in this paper are simple but corrected executed. However, there is an argument to be made that with such comprehensive data and well-supported knowledge on drivers of occurrence, more nuanced techniques could be used to explore the research question further (e.g. boosted regression trees are especially suited for interrogating covariate interactions – given that interacting conditions are found to be very important, I think you should consider trying such models).

Furthermore, the study tries to interrogate landscape-scale drivers of species occurrence with broad WorldClim climatic variables. This is difficult, because climate effects are often more important/easily-observed on larger scales – one would imagine that monthly or even weekly temperature and precipitation averages may have a more pronounced effect on this scale. Even better, soil moisture instead of precipitation, and fine-grained terrain relief data, may also prove to be important to the species, based on previous research highlighted in this manuscript. If possible, the authors should consider using these variables instead of WorldClim.

I find the conclusions in this study not fully supported by its results – the authors discussed management implications based on previous research and knowledge around the species, but the link between those discussions and the result in this study requires contextual and confirmatory knowledge of the species’ ecology. I strongly suggest including additional results, such as response curves and prediction maps, to further explain what precise environmental conditions are predicted to be favourable by the model.

Please find below a list of specific comments, in relation to line #

#44 this sentence can be shortened and merged with the previous

#86 considering the amount of available data per taxa, pathogenic species are not particularly under-represented in species distribution modelling, see e.g. 10.1016/j.fbr.2020.01.002 for a summary

#146-147 depending on the size of groves, within-grove climatic variations could be important and considered as a covariate

#150 it appears that ‘covariate’ is used to refer to factor variables in this manuscript, I suggest using only ‘covariate’ or ‘variable’ to refer to all predictors, as commonly practised by the species distribution modelling community

#188-189 are the data partitions sampled with replacement? Why not a standard 10-fold CV (dividing data into 10 distinct blocks and using 9 to train and 1 to test)

#231 I think these supplementary figures help readers a lot – mapped rasters of all continuous predictors can also be included in the Appendix, to better contextualise the environment in the study area

6. PLOS authors have the option to publish the peer review history of their article (what does this mean?). If published, this will include your full peer review and any attached files.

Reviewer #1: No

Reviewer #2: No

---

## [Author Response · Author response to Decision Letter 0]

21 Oct 2020

Please, see the attached response letter

---

## [Decision Letter · Decision Letter 1]

23 Nov 2020

PONE-D-20-11197R1

Climatic drivers of Verticillium dahliae occurrence in Mediterranean olive-growing areas of southern Spain

PLOS ONE

Dear Dr. Rodríguez,

Thank you for submitting your manuscript to PLOS ONE. After careful consideration, we feel that it has merit but does not fully meet PLOS ONE’s publication criteria as it currently stands. Therefore, we invite you to submit a revised version of the manuscript that addresses the points raised during the review process.

One of the referees still highlighted a couple of minor amendments that need to be done in the manuscript. I am more than confident you will handle them very quickly.

We look forward to receiving your revised manuscript.

Kind regards,

Mirko Di Febbraro

Academic Editor

PLOS ONE

Reviewers' comments:

Reviewer's Responses to Questions

**Comments to the Author**

1. If the authors have adequately addressed your comments raised in a previous round of review and you feel that this manuscript is now acceptable for publication, you may indicate that here to bypass the “Comments to the Author” section, enter your conflict of interest statement in the “Confidential to Editor” section, and submit your "Accept" recommendation.

Reviewer #1: All comments have been addressed

Reviewer #2: (No Response)

2. Is the manuscript technically sound, and do the data support the conclusions?

Reviewer #1: Yes

Reviewer #2: Yes

3. Has the statistical analysis been performed appropriately and rigorously? 

Reviewer #1: Yes

Reviewer #2: Yes

4. Have the authors made all data underlying the findings in their manuscript fully available?

Reviewer #1: Yes

Reviewer #2: Yes

5. Is the manuscript presented in an intelligible fashion and written in standard English?

Reviewer #1: Yes

Reviewer #2: Yes

6. Review Comments to the Author

Reviewer #1: I congratulate with the authors for the excellent work. All my comments have been successfully addressed.

Reviewer #2: Dear authors,

Thank you for the opportunity to re-review the manuscript titled ‘Climatic drivers of Verticillium dahliae occurrence in Mediterranean olive-growing areas

of southern Spain’. I am satisfied with the authors’ responses to the recommendations I raised in the first round. I believe this manuscript is now robust and valuable, with only a few minor tweaks, it can make a nice addition to the journal.

Although the changes from the previous version are not substantial, I am convinced by the authors’ responses to my previous comments, which demonstrated their comprehensive knowledge of the study system and I can now better appreciate the reasonings behind their conclusions. Below I detail my updated views on the concerns I raised in my previous review.

Regarding trialling more complicated modelling method: I am impressed that the authors fitted BRTs with multiple sets of parameters, and tested model performance rigorously. I think this result (that GLMs beat BRTs in performance) is worth mentioning in the text, or at least in an appendix, because other model-focused readers may also be interested in whether different modelling methods are tried for this research problem.

Regarding choice of covariates: I am convinced by the authors’ response that the covariates tried here are most suited to the study. Although I am still wondering whether seasonal and/or monthly climate data should be tested, since the authors identified that warm and wet conditions are most imported to V.dahliae, so maybe the specific susceptible month(s) are worth focusing on.

Regarding better contextualising the favourable conditions to V.dahliae: I think the new appendix maps of predictors will suffice, and I agree with the authors that occurrence prediction maps are not needed for this study.

I have a few other minor suggestions:

In table 1, I am not sure whether the hypothesis column is necessary, I believe the information is better conveyed in text, through a more detailed elaboration on how the top models supported the specific hypothesis.

Line 245: I suggest “compound” or “composite” over “sophisticated”, but I agree with the “primary” wording.

Line 286-290: I think the observed effect of plant nursery origin in the model deserves more discussion. Although it is beyond the scope of this study, it hints that non-climatic factors are also significant in moderating disease risk, this is worthwhile knowing from a management perspective.

Appendix S2: the colour scale on the last figure is confusing (it looks intuitively like a value that ranges from 0 to 1), I suggest changing it.

7. PLOS authors have the option to publish the peer review history of their article (what does this mean?). If published, this will include your full peer review and any attached files.

Reviewer #1: No

Reviewer #2: No

---

## [Author Response · Author response to Decision Letter 1]

4 Dec 2020

Please, see the attached response letter

---

## [Editor Report · Decision Letter 2]

14 Dec 2020

Climatic drivers of Verticillium dahliae occurrence in Mediterranean olive-growing areas of southern Spain

PONE-D-20-11197R2

Dear Dr. Rodríguez,

We’re pleased to inform you that your manuscript has been judged scientifically suitable for publication and will be formally accepted for publication once it meets all outstanding technical requirements.

Kind regards,

Mirko Di Febbraro

Academic Editor

PLOS ONE
---

## [Editor Report · Acceptance letter]

17 Dec 2020

PONE-D-20-11197R2 

Climatic drivers of *Verticillium dahliae* occurrence in Mediterranean olive-growing areas of southern Spain   

Dear Dr. Rodríguez:

I'm pleased to inform you that your manuscript has been deemed suitable for publication in PLOS ONE. Congratulations! Your manuscript is now with our production department. 

Kind regards, 

on behalf of

Dr. Mirko Di Febbraro 

Academic Editor

PLOS ONE